# Vector borne disease control interventions in agricultural and irrigation areas in sub-Saharan Africa: A systematic review

**Levi Kalitsilo**[1]*, **Leila Abdullahi**[2], **Nyanyiwe Mbeye**[3], **Lily Mwandira**[4], **Hleziwe Hara**[1], **Collins Mitambo**[1], **Rose Oronje**[2]

1 African Institute for Development Policy, Lilongwe, Malawi, 2 African Institute for Development Policy, Nairobi, Kenya, 3 Kamuzu University of Health Sciences, Blantyre, Malawi, 4 Clinton Health Initiative, Lilongwe, Malawi

* levi.kalitsilo@afidep.org

**Data Availability Statement:** All relevant data are within the manuscript and its Supporting information files.

## Abstract

Irrigation farming has raised concerns about the steady transmission and introduction of new vector-borne infectious diseases (VBD) in the areas involved. This systematic review aimed to determine interventions that are effective for the management and control of VBDs in irrigation areas in sub-Saharan Africa (SSA). We searched the literature on VBD interventions in SSA from published and grey literature without specifying the publication year. A search strategy identified 7768 records from various databases, and after screening, 16 were included in the final analysis. Results showed various VBD control interventions were effective, including indoor residue spray (IRS), insect-treated nets (ITN), larva source management (LSM), mass drug administration (MDA), integrated vector management (IVM), and molluscididing. IVM was commonly practiced, and its success was because of the complementarity of the various interventions involved. Successful VBD control interventions led to improved health amongst irrigation communities and consequently improved agricultural productivity. However, some challenges to these interventions were identified, which include seasonal changes and climate variability, insecticide and drug resistance, and farmers' attitudes toward accepting the interventions. Regardless, results showed that VBD control and management can be integrated into irrigation farming before or after the establishment of the irrigation scheme.

## 1. Background

Vector-borne diseases (VBDs) are human illnesses caused by parasites, viruses, and bacteria and are transmitted by living agents such as mosquitoes, snails, ticks, tsetse flies, lice, etc. [1]. Some of the major VBDs in Africa include malaria, yellow fever, dengue, schistosomiasis, onchocerciasis, and human African trypanosomiasis (HAT) [1]. Globally, it is estimated that the major VBDs account for 17% of the global burden of all infectious diseases and cause over 700,000 deaths annually [2, 3]. The most prevalent VBD is malaria, and in 2021, the World Health Organisation (WHO) estimated there were 247 million cases and 619,000 malaria-

**Funding:** NIHR funded the study through the Shire-Vec Project.

**Competing interests:** The authors have declared that no competing interests exist.

related deaths globally [4]. Although other VBDs such as schistosomiasis, lymphatic filariasis, HAT, and onchocerciasis are less deadly, they still result in high levels of morbidity and are of local importance in specific areas of populations [1, 2]. Further, the huge burden of morbidity and mortality worldwide arising from the VBDs particularly affects the poorest of the poor [5], and this is disproportionate in tropical and sub-tropical areas, with most deaths occurring in children under the age of 5 [3, 6]. VBDs are widespread throughout sub-Saharan Africa (SSA), and many are co-endemic [1].

Climate change and the growing population seem to have increased demand for food and energy the world over, resulting in countries having to modify the environment by constructing dams and irrigation schemes in response to the growing need. However, the dams and irrigation schemes have raised concerns about the steady transmission, and amplification, or introduction of new vector-borne infectious diseases in the areas involved [7]. Particularly, there is a sustained and widespread impact of agricultural conversion on VBD prevalence in that the expansion or redistribution of freshwater via irrigation or dams creates new habitats for vectors that have aquatic life-cycle stages [8, 9]. Furthermore, changes in land use driven by irrigation schemes alter the interaction and abundance of vectors and people and potentially lead to increased opportunities for exposure. For example, living near large-scale irrigation dams has been associated with increased risk for malaria [8] and schistosomiasis transmission [9], with permanent aquatic habitats extending the transmission seasons and geographical range of these diseases.

Countries and international organisations (including WHO) are making efforts to control and manage VBDs. However, there is notably inadequate delivery of vector control interventions because of limited resources in many countries, and this is partially responsible for the high burden of risk for VBDs [2]. The climatic and socio-economic conditions in Africa make the continent the most vulnerable to VBDs, which have been noted to exacerbate the level of poverty and exert an immense toll on the economies of countries [2]. The VBDs, therefore, usually affect the poorest populations, particularly where there is a lack of access to adequate housing, safe drinking water, and sanitation [10]. A reduction in the VBD burden will thus reduce household poverty, enable greater growth and productivity, strengthen health systems, and increase equity and empowerment for women [6]. This systematic review focused on the integration of various interventions to address VBDs in irrigation and agricultural systems in SSA.

## 1.1 Interventions to address VBDs

Vectors can transmit infectious diseases between humans or from animals to humans. An effective approach against most of the important VBDs is targeting the vectors that transmit disease-causing pathogens. Reducing vector survival and human-vector contact suppresses and halts transmission and is often the best-proven or only available preventive measure against most VBDs, yielding one of the highest returns on investment in public health [2]. Many vectors transmitting VBDs are bloodsucking insects that ingest disease-producing microorganisms during a blood meal from an infected host (human or animal), and later on, they inject them into new hosts during their next blood meal [11]. For example, people get malaria when bitten by infected Anopheles mosquitoes; the Aedes mosquito is the primary vector of dengue and also largely attributed to the transmission of yellow fever and the chikungunya virus, particularly in Africa; Anopheles and Aedes mosquitoes transmit lymphatic filariasis; sandflies transmit leishmaniasis; onchocerciasis is transmitted through the bite of blackfly; and freshwater snails transmit schistosomiasis [1].

**Table 1. Vector control interventions, targeted vectors and vector-borne diseases.**

| | Interventions | | Targeted vectors | VBDs |
|---|---|---|---|---|
| Chemicals & Pesticides | Indoor residual spraying (IRS) | | Mosquitoes, | Malaria; Lymphatic filariasis; Dengue; Leishmaniasis; Chikungunya; Schistosomiasis |
| | Long-Lasting Insecticidal Nets (LLINs) | | Mosquitoes, ticks | Malaria; Lymphatic filariasis; Dengue; Leishmaniasis; Chikungunya |
| | Outdoor spraying | | Blackflies, Mosquitoes | Malaria; Lymphatic filariasis; Onchocerciasis; dengue and Yellow fever |
| | Traps and targets (insecticide-treated) | | Tsetse flies | HAT |
| | Molluscicides | | Snails | Schistosomiasis |
| | Larval source management | Larviciding | Mosquitoes | Malaria; Dengue; Leishmaniasis; Onchocerciasis (chemical or microbial larvicides); Yellow fever; West Nile virus; Chikungunya. |
| | | Habitat modification and manipulation/ Environmental management | Mosquitoes, Snails, Black flies, Tsetse flies | Lymphatic filariasis; Dengue; Leishmaniasis; Schistosomiasis; |
| | | Biological control | Mosquitoes, Snails | Lymphatic filariasis; Dengue; Chikungunya; Schistosomiasis (with fish); |
| | | Polystyrene beads | Mosquitoes | Lymphatic filariasis |
| | Medication | | | Lymphatic filariasis; Schistosomiasis (preventive chemotherapy); Onchocerciasis (ivermectin chemotherapy for parasite); Malaria (chemoprevention through mass drug administration; vaccine anti-malarial drugs) |
| | Personal protection | | Mosquitoes | Malaria |
| | Social mobilization campaigns (community education & public relations) including practical field-based education | | | Dengue; Schistosomiasis |

## 1.2 How the vector control interventions work in managing VBDs

There are several vector control methods and interventions, some targeting specific vectors and others targeting two or more vectors, as shown in Table 1 below.

WHO recommends the use of insecticides in water bodies as a supplementary intervention to ITNs or IRS for malaria prevention [6]. Integrated Vector Management (IVM) is a cross-cutting intervention that involves using a range of proven vector control methods, either in combination or alone; several methods against a single disease; or single or several methods against several diseases [1]. This approach aims to make vector control more cost-effective, efficient, ecologically sound, and sustainable [1]. The WHO promotes IVM to control one or more VBDs using multiple interventions [1, 2]. IVM differs from routine vector control in that it incorporates interventions, actors, and resources that are coordinated between health and other sectors (non-health ministries such as agriculture and forestry, the private sector, and communities) [1].

## 1.3 Rationale for the review

This review explored the applicability of integrating various VBD control interventions into irrigation schemes and systems. Interventions for vector control have been noted to yield one of the highest returns on investment in public health [2]. A study on VBDs and associated factors in Ethiopia established that controlling and managing VBDs requires designing appropriate intervention strategies specific to communities [10]. In Zimbabwe, a study on IVM for improved disease prevention showed that most VBD control programmes target a single disease and recommended IVM for most effective vector control and resource use efficiency [12]. While most previous studies have largely focused on vector control for malaria, this study synthesised evidence that included other vectors such as snails and blackflies. Further, the

impact of irrigation on VBDs depends on the geographical and climatological context. Different sectors implementing multiple approaches are required to control and eliminate VBDs [2]. The study synthesised various vector control interventions to make a case for the need for effective VBD control methods to be implemented and integrated into irrigation schemes and systems that can be adapted to specific geographical and climatological contexts.

### 1.4 Objectives

1. To determine interventions for the management and/or control of vector-borne diseases in agricultural and irrigation areas in SSA.

2. To document challenges and lessons on management and control of VBDs in agricultural irrigation schemes in SSA.

## 2. Methods

### 2.1 Criteria for considering studies for this review

**2.1.1 Types of studies.** The review included observational study designs such as cross-sectional, case-control, and cohort studies, as well as intervention studies including quasi-randomized studies, cluster randomised controlled trials, and individual randomised controlled trials (RCTs).

**2.1.2 Types of participants.** The review involved studies with populations of all ages living in close proximity to agricultural and irrigation schemes or projects.

**2.1.3 Types of interventions.** The interventions included single or a combination of the following:

a. Insecticide Treated Nets (ITNs)

b. Indoor Residual Spraying (IRS)

c. Larval source management (LSM)

d. Insecticide-treated traps and targets (ITTT)

e. Personal protection measures with a primary use-pattern to protect individual users

f. Medication

g. Mass drug administration (MDA)

h. Mollusciciding

i. Social mobilization campaigns for behavioural change.

j. Integrated vector management (IVM)

**2.1.4 Types of comparison.** Normal practice, alternative intervention(s), or no intervention(s).

**2.1.5 Types of outcome measures.** Primary outcomes

- Reduction in prevalence of VBDs (i.e. malaria, schistosomiasis, yellow fever, dengue, human African trypanosomiasis (HAT), lymphatic filariasis, leishmaniasis).

- Behavioural changes, including the adoption of interventions by farmers, community members, irrigation scheme implementers, and countries.

Secondary outcomes

- Adverse events following interventions

- Cost-effectiveness/cost of interventions

- Change/amendments in policies or strategies following interventions to address VBDs

## 2.2 Search methods for identification of studies

We developed a comprehensive search strategy for peer-reviewed studies and grey literature with no language and time limit. Comprehensive search terms with MESH subheadings using key terms like control, mosquitoes, malaria, zika, yellow fever, irrigation, and Africa. Using the search strategy in Table 2 below, we searched electronic databases including PubMed, Cochrane Library, Trip database, African Index Medicus, WHO database, and Google Scholar for both peer-reviewed and grey literature. In addition, the review team screened the reference lists of all the included studies and related systematic reviews for other potentially eligible primary studies. The review protocol was registered in the International Prospective Register of Systematic Reviews (PROSPERO), registration number CRD42022376242.

**Table 2. Search terms.**

|      | Themes | Search Terms |
| --- | --- | --- |
| #1P | Population in irrigation areas | All ages human population |
| #2S | Settings in irrigation areas | Irrigation OR Farming OR Dam OR Wetlands OR Agricultural areas |
| #3I | Indoor residual spraying, ITNs, integrated vector management, etc. | IRS OR Indoor Residual Spray* OR larvicid* OR ITNs OR Insecticide Treated Nets OR Long-lasting insecticide-treated nets OR LLINs OR Nets OR Molluscicides OR Mass drug administration OR Health education OR Community awareness OR Community sensitization OR Training OR education OR Integrated Vector Management OR Pesticides OR Vector control |
| #4C | Standards practice/routine practice | Routine OR Standard practice OR nothing |
| #5O | Prevalence of VBDs Behavioral changes | malaria OR malaria transmission OR Plasmodium OR onchocerciasis OR schistosomiasis OR lymphatic filariasis OR dengue OR yellow fever OR chikungunya OR zika virus OR leishmaniasis OR sleep* sickness OR African trypanosomiasis OR Malaria OR Bilharzia OR Schistosomiasis OR Dengue OR onchocerciasis OR Chikungunya OR Yellow fever OR African trypanosomiasis OR sleep* sickness OR leishmaniasis OR lymphatic filariasis OR Vector borne diseases |
| #6 | SSA | Algeria OR Angola OR Benin OR Botswana OR "Burkina Faso" OR Burundi OR Cameroon OR "Cape Verde" OR "Cabo Verde" OR "Central African Republic" OR Chad OR Comoros OR Comores OR Comoro OR Congo OR "Congo-Brazzaville" OR "Congo Republic" OR "Republic of the Congo" "Côte d'Ivoire" OR "Democratic Republic of the Congo" OR "DR Congo" OR DRC OR "Congo-Kinshasa" OR Djibouti OR "Equatorial Guinea" OR Eritrea OR Ethiopia OR Gabon OR Gambia OR "The Gambia" OR Ghana OR Guinea OR Guinea-Bissau OR Kenya OR Lesotho OR Liberia OR Madagascar OR Malawi OR Mali OR Mauritania OR Mauritius OR Mozambique OR Namibia OR Niger OR Nigeria OR Rwanda OR "Sao Tome and Principe" OR "São Tomé and Príncipe" OR Senegal OR Seychelles OR "Sierra Leone" OR Somalia OR "South Africa" OR "South Sudan" OR Sudan OR Swaziland OR Togo OR Uganda OR "United Republic of Tanzania" OR Tanzania OR Zambia OR Zimbabwe |
|      | Combined | #1 AND #2 AND #3 AND #4 AND #5 AND #6 |

**2.2.1 Selection of studies.** The first electronic database search was conducted on December 12th, 2022, and updated on October 29th, 2023. Two review authors (LK and CM) independently screened the titles and abstracts of each identified study. All irrelevant topics, duplicates, and studies that did not meet the inclusion criteria were excluded. Thereafter, the authors (LK and CM) screened the full texts of each potentially relevant study based on the inclusion criteria. Any disagreements between the two review authors were resolved by discussion and consensus. Where no agreement was reached, a third review author was asked to arbitrate (LA or NM). Multiple publications of the same study were removed.

**2.2.2 Data extraction and management.** Two review authors (LK and CM) independently extracted relevant information from all identified articles that met the inclusion criteria. The data extracted included study identity, design, country, participants, aim, interventions, and outcome measures. Where no agreement was reached, the third or fourth review author was asked to arbitrate (LA or NM).

## 2.3 Assessment of risk of bias in included studies

Each of the included studies was assessed for risk of bias using the Critical Appraisal Skills Programme (CASP) risk of bias tool for various studies. CASP has appraisal checklists designed for use with cohort studies [13], case-control studies [14] and qualitative studies [15]. The other appraisal tools used were a cross-sectional study tool [16] and a mixed method risk assessment tool [17].

**2.3.1 Dealing with missing data.** There was no missing data situation in the studies included in the final analysis.

**2.3.2 Assessment of reporting biases.** Test for asymmetry with a funnel plot was not applicable as the data was analysed in a narrative format.

## 2.4 Data synthesis

The data was qualitatively synthesised from studies of similar study designs, similar interventions, similar participants, and similar outcomes. Overall, we interpreted the study findings by considering the methodological quality of the studies and the strength of the evidence. For each observed effect, we explicitly stated the strength of the evidence and drew conclusions. The qualitative data collected was independently analysed by two team members (LK and CM) and verified by another team member (LA or NM). The analysis was done manually.

**2.4.1 Subgroup and sensitivity analysis and investigation of heterogeneity.** The subgroups were made according to study design, disease, or intervention and this ensured meaningful investigation of heterogeneity.

## 2.5 Quality of evidence

The review team planned to use the GRADE approach to assess the certainty of evidence at the outcome level [18]. However, there was insufficient information from the included studies to assess the quality of evidence.

## 3. Results

We searched for studies that were carried out in SSA, whose population was humans, and the VBD control intervention (s) impacted VBD prevalence in the area. A total of 7,768 records were identified by the search strategy from various databases, published literature (n = 2,915), and grey literature (n = 4,853). After excluding 889 duplicates, we screened 6,879 records and found that 6,800 records were not relevant to our review question. The team reviewed the

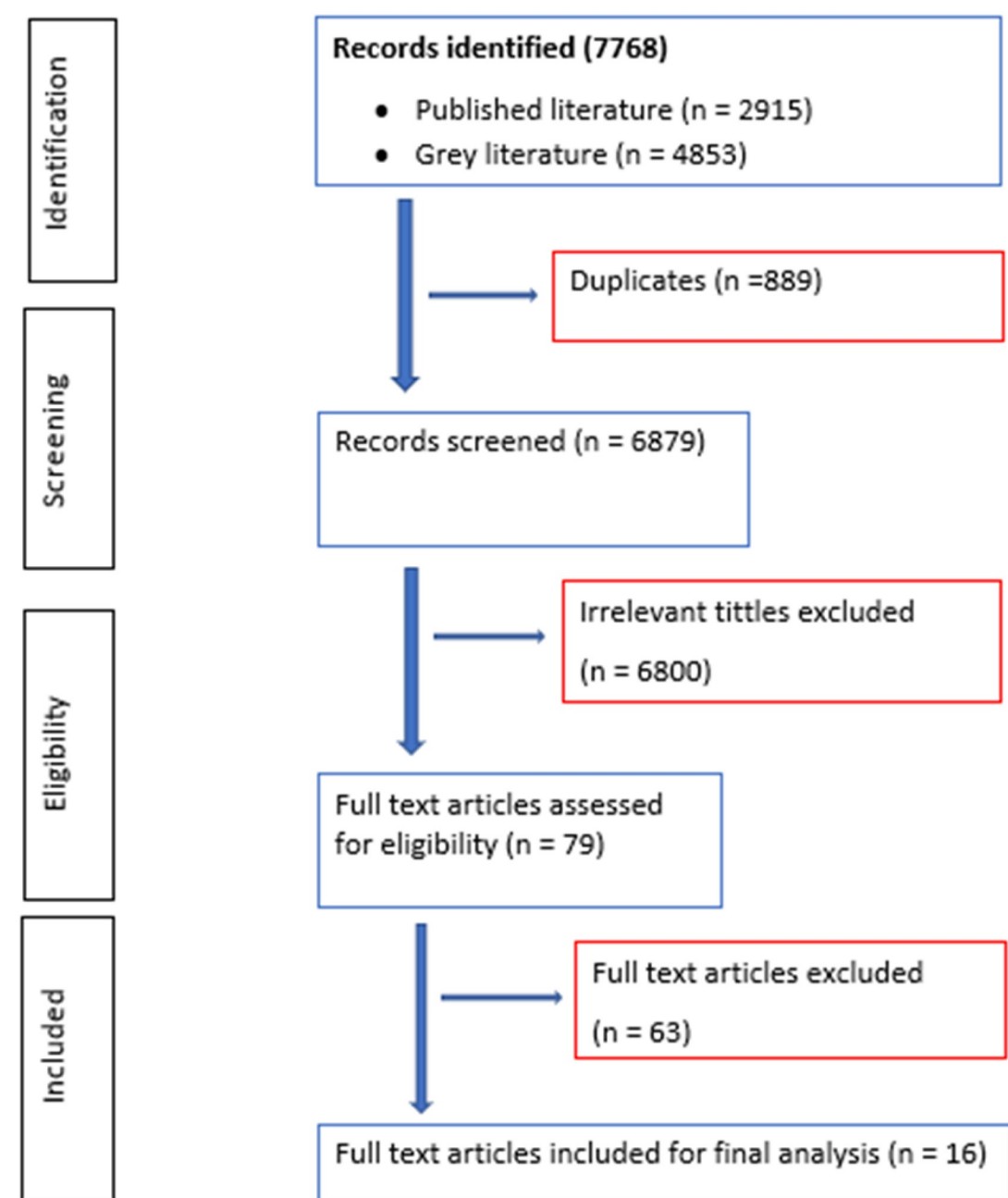

**Fig 1. PRISMA diagram showing the article selection process.**

remaining 79 potentially eligible full-text articles for inclusion and excluded 63 of them for the reasons given in S1 Appendix. Therefore, 16 studies were included in the final analysis.

The study selection process is illustrated in the diagram below (Fig 1).

For full search strategies for each database, see S2 Appendix.

## 3.1 Characteristics of the included studies

**3.1.1 Study designs.** Amongst the included studies, 4 were cross-sectional studies (n = 4) [19–22]; 3 experimental trials (n = 3) [23–25]; 1 controlled before and after study (n = 1) [26,

27]; 1 randomised cross-over trial (n = 1) [28]; 6 mixed methods studies (n = 6) [26], [29–33]; and a cohort study (n = 1) [34]. In total, there were 16 included studies.

**3.1.2 Study settings.** All included studies were from the SSA region in areas where modern and traditional irrigation farming is practiced, as shown in Table 3. A majority were from Ethiopia (n = 3) [20, 24, 31]; then Kenya (n = 3) [21, 27, 29, 33]; Sudan (n = 2) [22], [32]; The Gambia (n = 1) [28]; Zimbabwe (n = 1) [23]; Uganda (n = 1) [30]; Senegal (n = 1) [26]; Tanzania (n = 1) [25]; Chad (n = 1) [35]; Burkina Faso (n = 1) [34] and; Malawi (n = 1) [19].

**3.1.3 Study population.** The studies involved different VBD control interventions that targeted specific or all age groups in the population. Seven studies targeted all ages [19, 21, 25, 26, 31, 35, 36]; one targeted children-aged between 8 and 14 [30]; three targeted school aged children [20, 29, 32]; two targeted children between 0.5 and 10 years [27, 28]; one study targeted children under 10 years of age [24]; one study targeted children under 6 years of age [23]; and another one targeted children under 5 years of age [22].

**3.1.4 VBD control interventions.** The included studies reported various VBD control interventions, including IRS (n = 1) [29]; LLIN (n = 1) [19]; Larval Source Management (n = 4) [20, 24, 28, 31]; mollusciciding (n = 1) [23]; biological control (n = 1) [26]; social mobilisation (n = 1) [30]; Mass Drug Administration (MDA) (n = 2) [33, 34]; and IVM, involving a combination of two or more interventions (n = 5) [21, 22, 25, 27, 32]. Out of the 16 studies, nine focused on malaria with mosquitoes as targeted vectors, six on schistosomiasis with snails as the targeted vectors, and one on onchocerciasis targeting blackflies. There were no studies for the other VBDs that fit the selection criteria.

A total of five studies were reported on IVM. Of these, three focused on malaria with various combinations of interventions: one specifically on the combination of IRS, LLIN, antimalarial treatment, and malaria diagnosis [22]; another on the combination of ITN, environmental management, and anti-malarial treatment [21]; and one on the combination of ITN and LSM [27]. Two were about schistosomiasis, with one combining health education (water sanitation and hygiene—WASH) with snail control and chemotherapy [32], and the other combining mollusciciding with mass drug administration [25].

Four studies reported on LSM with malaria as the targeted VBD. Two reported on water management, specifically, one was on canal water management [31] and the other on varying irrigation systems [20]. For the other two; one was on micro-larviciding [28] and the other one on environmental management [24].

Two studies reported on MDA. One targeted schistosomiasis with anthelmintics [33] and another targeted onchocerciasis with ivermectin [34].

One study reported on IRS [29] targeting malaria and another on LLIN [19]. Another study reported on mollusciciding [23], targeting schistosomiasis, one on biological control [26] targeting schistosomiasis and the other study reported on social behaviour change [30]. Table 3 below summarises the characteristics of the included studies.

A summary characteristic of excluded articles is presented in S1 Appendix.

## 3.2 Primary outcomes

**3.2.1 Reduction in VBDs.** Five studies reported on a combination of two or more VBD control interventions, also known as IVM. Among them, three focused on malaria and two on schistosomiasis. Overall, IVM in irrigation areas reported remarkable success in VBD control and management. A study by Okech et al. focusing on malaria in the Mwea irrigation settlement in Kenya and combining ITN, environmental management, and antimalarial treatment interventions reported a reduction in clinical malaria cases from about 40% before IVM to zero at the end of the study [21]. Another study by Elmardi et al. in the Gezira and

**Table 3. Summary characteristics of included studies.**

| ID | Study type | Intervention | Control | VBD | Vector | Population/ Age | Country | Duration of intervention | Outcomes |
|---|---|---|---|---|---|---|---|---|---|
| Zhou G et al., 2022 | Mixed methods | IRS | Non-intervention | Malaria | *Anopheles gambiae sensu lato* and *Anopheles funestus sensu lato* | School going children | Kenya | 5 years | IRS interventions were followed by a decline in plasmodium infection prevalence. |
| Mangani C. et al., 2021 | Cross-sectional | LLIN | Non-irrigated area | Malaria | *Anopheles arabiensis* and *An. funestus* | All age groups | Malawi | 1 year | Participants who reported not using an LLIN the previous night had a higher prevalence of malaria infection compared with those who reported using LLINs (34.6% versus 28.1%). |
| Jaleta KT et al., 2013 | Cross-sectional | LSM: Irrigation farming/Water management | Non-irrigated agriculture or rain-fed agriculture | Malaria | *An. arabiensis* | School-aged children aged from 4–15 years | Ethiopia | 1 year | In rain-fed agro-ecosystem villages, annual malaria prevalence (2.9%) was significantly lower (P <0.05) than in the two villages practising irrigation |
| Okech BA et al., 2008 | Cross-sectional | IVM: ITN, environmental management and antimalarial treatments | No intervention | Malaria | *An. arabiensis* and *An. funestus* | All ages (389) and then school-aged children (300) | Kenya | 8 months | Malaria cases in the community hospital reduced from about 40% in 2000 to less than 10% by 2004 and by the year 2007, malaria cases decreased to zero. |
| Elmardi KA et al., 2021 | Cross-sectional | IVM: utilization of malaria diagnosis, utilization of Artemisinin-based combination therapy (ACT), IRS and LLIN | No intervention | Malaria | *An. arabiensis* | Children under 5 years of age | Sudan | 5 years | IRS coverage was associated with malaria infection (Odds ratio 0.83 per 10% coverage, 95% Confidence Interval (95%CI) 0.74–0.94, p = 0.003) indicating that a higher level of IRS coverage was associated with less malaria infection. |
| Kibret S et al., 2014 | Mixed methods | LSM: Water Management/ Canal water management | Non-irrigated area | Malaria | *An. arabiensis, An. pharoensis and An. coustani.* | All age groups | Ethiopia | 1 year | Overall, monthly malaria incidence was over six-fold higher in the irrigated villages (33.7%) than the non-irrigated (5.6%) villages during the study period. |

(*Continued*)

**Table 3.** (Continued)

| ID | Study type | Intervention | Control | VBD | Vector | Population/Age | Country | Duration of intervention | Outcomes |
|---|---|---|---|---|---|---|---|---|---|
| Yohannes M et al., 2005 | Experimental | Larva Source reduction: Environmental management | Non-irrigated area | Malaria | *An. arabiensis* | Children under 10 years of age | Ethiopia | 10 months | Environmental management was able to reduce the abundance of malaria vectors with minimum participation of the community and by utilizing local resources bringing down the prevalence. |
| Fillinger U et al., 2009 | Controlled before and after | IVM (ITN and LSM) | Non-larva control intervention groups | Malaria | Anopheles | Children between 0.5 to 10 years | Kenya | 3 years | The risk of acquiring new parasite infections in children was substantially and independently reduced by ITN use (odds ratio, OR: 0.69; 95% confidence interval, CI: 0.48–0.99) and larvicide application (OR: 0.44; 95% CI: 0.23–0.82), after adjusting for confounders. |
| Majembere S et al., 2010 | Randomised cross over | Larval Source Management | Non-intervention habitats | Malaria | *An. gambiae s.l.* | Children between 0.5 to 10 years | Gambia | 2 years | There was no reduction in clinical malaria or anemia. The study demonstrated that larviciding reduced the aquatic stages of anophelines, but only limited success in reducing adult numbers in three of the four study zones. |
| Shiff CJ et al., 1973 | Experimental trial | Molluscicide | No molluscicide | Schisto | *Biomphalaria pfeifferi* and *Biomphalaria bulinus* | Children under 6 years of age | Zimbabwe | 4 years | Transmission of both schistosoma haema-tobium and S. mansoni reduced to levels below anywhere in the country even where irrigation is not practiced |
| Tameim O et al., 1985 | Mixed methods | IVM: Health education (WASH), snail control and chemotherapy | No intervention | Schisto | B. pfeifferi | School going children | Sudan | 4 years | Transmission reduced from 14% to less than 10% |
| Trienekens SCM et al., 2022 | Mixed methods: Longitudinal and cohort | Behaviour/Social mobilisation | No intervention | Schisto | *Biomphalaria choanomphala*, *Biomphalaria sudanica* and *B. pfeifferi* | Children aged 8 to 14 | Uganda | 1 year | Infection decreased from 50% to 18.75% |

(*Continued*)

**Table 3.** (Continued)

| ID | Study type | Intervention | Control | VBD | Vector | Population/ Age | Country | Duration of intervention | Outcomes |
|---|---|---|---|---|---|---|---|---|---|
| Fenwick A and Jorgensen T. A., 1972 | Experimental trial | IVM: Mollusciding and Mass treatment | No intervention | Schisto | *B. pfeifferi* | All age groups | Tanzania | 2 years | Infection reduced by 50% from 44% prevalence to 22%. |
| Sokolow s. et al. 2015 | Mixed methods | Biological control/ restoring native river prawns | No intervention | Schisto | *B. pfeifferi* and *B. bulinus* | All ages | Senegal | 1.5 years | S. hematobium prevalence and infection intensity changed from 64% to 58% and 11% to 6% respectively in 18 months. Human schistosomiasis prevalence was18 ±5% lower and egg burden was 50±8% lower at the prawn-stocking village compared with the control village. |
| Lelo A et al., 2014 | Mixed methods | Mass Drug Administration (MDA) | 52 non-phenotypically susceptible children | Schisto | *B. pfeifferi* | Students between 5 and 14 years of age | Kenya | 4 years | Found no evidence of reduced transmission or schistosome population decline over the course of the program |
| Nikiema A S et al. | Cohort study | Mass drug administration (MDA): ivermectin | No intervention | Oncho | Blackflies of the genus *Simulium* (*Diptera*: *Simuliidae*) | Aged between 5 and 70 years | Burkina Faso | 1 year | The percentage reduction of the mean microfilarial load varied from 87.1% to 97.9% with an overall mean of 92%. |

Sennarirrigation schemes of Sudan, which combined the utilisation of malaria diagnosis and the utilisation of artemisinin-based combination therapy (ACT), IRS, and LLIN, also reported a reduction in malaria infection. A comparison of individual interventions within this IVM found that IRS coverage brought the most effectiveness, with malaria cases reportedly getting lower as coverage was increased (OR: 0.83 per 10% coverage; CI 0.74–0.94; p = 0.003) [22]. In a study that combined ITN and LSM in the Papyrus swamps of Western Kenya by Fillinger et al., the interventions reportedly reduced malaria infection risk from 32% to 7%. Assessing individual intervention effectiveness showed that ITN substantially achieved a reduction in the risk of acquiring new parasite infections in children (OR: 0.69; CI: 0.48–0.99), and the addition of LSM enhanced the achievement significantly (OR: 0.44; CI: 0.23–0.82) [27]. An IVM study focusing on Schistosomiasis in the Rahad irrigation scheme in Sudan, combining water, sanitation, and hygiene (WASH) practices, snail control, and chemotherapy interventions, reported reduced transmission from 14.4% to less than 9.3% [32].

Four studies reported on LSM with varied degrees of success. A study comparing three farming systems that had different water management systems in Ethiopia (all-year sugarcane irrigation, traditional irrigation, and no irrigation) found that availability of water throughout the year in sugarcane irrigation schemes resulted in very high vector abundance and an annual malaria prevalence of over 7.2%; in traditional irrigation scheme, vector abundance was high

with an annual malaria prevalence of 5.3%; and where no irrigation was practiced, vector abundance was very low with a malaria prevalence of only 2.9%. Remarkably, no malaria cases were reported during the dry season in the non-irrigated area [20]. In Ziway (Ethiopia), a study reported a proliferation of malaria vector mosquito breeding sites due to poor canal water management (unregulated water releases), thereby intensifying malaria transmission in the irrigated communities. Comparatively, irrigated areas recorded six times higher malaria incidences (33.7%) than the non-irrigated (5.6%) communities [31].

LSM by managing the environment to eradicate predominant breeding sites of Anopheles gambiae s.l., such as seepage at the dam base, leaking irrigation canals, pools, that formed along the bed of streams from the dam, and man-made pools reduced vector abundance by 49% (95% CI: 46.6–50.0). However, very few malaria cases were reported during the study, making it difficult to ascertain the significance of the intervention on VBD control and management [24]. Similarly, along the banks of the Gambia River in West Africa, LSM reported success in reducing the aquatic stages of anophelines but, to a greater extent, showed limited success in reducing adult numbers and therefore recorded no significant reduction in malaria cases [28].

The use of IRS for VBD management and control in an irrigation area reported great success. A study comparing outcomes in irrigated and non-irrigated areas of Kisumu and Homa bay in Kenya indicated a risk ratio of 4.80 (CI: 1.45–26.94) and a p-value of 0.0099 in Kisumu irrigated area, which had no IRS; a risk ratio of 0.94 (CI: 0.85–1.05) and a p-value of 0.2722 in Homa bay, which was irrigated and had IRS; and a risk ratio of 0.70 (CI: 0.54–0.90) and a p-value of 0.0058 in another part of Homa bay, which was non-irrigated and had IRS. This meant that while IRS was effective in controlling malaria in Homa bay (both irrigated and non-irrigated), irrigation farming compromised the effectiveness of IRS to some extent [29].

Reporting on LLIN, one study in Malawi showed that populations living closer to an irrigation scheme had a significantly higher malaria infection rate than those living far away, 32.0% versus 26.6%, respectively. In ascertaining the effectiveness of LLIN in the same area where ownership was 69.7%, malaria infection was reportedly 34.6% amongst those who were not using LLIN despite having them, versus 28.1% amongst those who constantly used the LLIN [19].

Mass Drug Administration on different VBDs reported mixed results. While a study of ivermectin impact on onchocerciasis in Burkina Faso reported a very positive 92% average microfilarial load reduction [34], a four-year school-based MDA (anthelmintics) to control schistosomiasis in Mwea (Kenya) found no evidence of reduced transmission or schistosome population decline despite the intervention reducing schistosome infection morbidity [33]. However, the MDA in Kenya had strong benefits to individual health as fewer children were infected over time.

Mollusciciding registered success in the management and control of schistosomiasis. A study in Zimbabwe reported an over 50% reduced transmission of both Schistosoma haemalterbium and Schistosoma mansoni amongst people in the South Eastern Lowveld large-scale irrigation area from 8.72 to 3.6%, which was more significant than ever reported anywhere in the country [23]. A reduction in the disease prevalence among farm workers resulted in increased labour efficiency and improved productivity on the farms.

Behaviour change activities also recorded remarkable success in the control and management of schistosomiasis. Targeted behaviour changes on specific water contact practices that centred on water, sanitation, and hygiene (WASH) amongst school-going children in Uganda led to a decrease in schistosoma infection from 50% to 18.75% [30].

One study on biological control saw the re-introduction of native river prawns in Senegal, resulting in S. hematobium prevalence and infection intensity changing from 64% to 58% and

11% to 6%, respectively, within 18 months. Comparatively, human schistosomiasis prevalence was 18±5% lower, and egg burden was 50±8% lower in communities where the prawns were restocked compared to the control communities [26].

**3.2.2 Behavioural changes.** Four studies reported on community behavioural change as a result of VBD control and management interventions. During a four-year IVM study investigating the potential factors that could have contributed to the decline of malaria cases in a hospital in Mwea (Kenya), the intervention led to the adoption of positive behaviour by the communities. After noting the importance of malaria control, Mwea community members registered increased ITN usage and the use of other combinations of malaria control interventions [21]. Another study in Kenya by Fillinger et al. found that combining LSM and ITN led to positive behaviour, with observable ITN usage increasing in households from 4.8% at baseline to 40.8% during the study period [27]. Targeted behavioural change activities amongst school-going children in Uganda led to the adoption of positive WASH behaviours. Improved water contact practices led to improved individual health behaviour among school children resulting in a reduction in schistosome infection rates [30]. Another behaviour change outcome was noted in an MDA programme targeting school-going children in Uganda who, due to the intervention, adopted positive health behaviours, which led to a decline in schistosome infection over time [33].

## 3.3 Secondary outcomes

**3.3.1 Adverse effects.** No study reported any adverse effects as a result of any VBD control and management intervention.

**3.3.2 Cost-effectiveness of interventions.** Three studies reported on the VBD control intervention's cost-effectiveness. Two centered on LSM and one on IVM. While ITN and targeted IRS have been largely touted as cost-effective, a comparison between them and LSM in Western Kenya indicated that LSM is as cost-effective as targeted IRS and ITN, with indications that it may be even less costly with environmental modifications and targeted application [27]. An IVM study with a combination of health education (WASH), snail control, and chemotherapy to control schistosomiasis in the Rahad irrigation scheme in Sudan found that the complementarity of the interventions was more cost-effective than using molluscicides alone [32].

**3.3.3 VBD control policy.** One study reported on how VBD interventions influenced policy. The success of ivermectin in reducing onchocerciasis in Burkina Faso led to the government sustaining the biannual MDA since 2011 [34].

## 3.4 Challenges and lessons on management and control of VBDs

Despite registering great strides in vector control and VBD reduction, some interventions reported challenges and provided lessons for VBD management and control intervention improvement in irrigation areas. As observed in a study by Yohannes et al. on assessing whether environmental management could reduce malaria transmission, LSM in irrigation was notably challenging, especially where it covers very large, densely populated, and remote areas. Identifying and covering all potential breeding sites was hard to achieve. Further, seasonal changes and climate variability, which influence vector breeding patterns, may make it difficult for LSM over large areas [24]. A lesson from this is to have clear LSM intervention plans and involve local communities that are familiar with the geographical setting.

Another challenge was registered in the biological control of schistosomiasis. A study at the Diama Dam area in Senegal reported that biological conservation does not always benefit human health. Since river prawns are human food, the intervention may compete with

communities' interests as they would need the same for domestic consumption. However, a lesson from the same is to scale up river prawn restoration and aquaculture to ensure that they can replenish naturally and therefore provide food to the communities without affecting the critical prawn threshold required for snail elimination [26].

The last challenge was noted in MDAs. While acknowledging that MDA targeting school-going children only did not register expected outcomes in Mwea (Kenya), it was observed that extending the intervention to the larger community would reduce parasite transmission but increase the chances of drug resistance due to the selective pressure that the parasites may be subjected to [33]. This therefore calls for systematic targeting of the population in MDA interventions.

## 3.5 Risk of bias in included studies

The included studies were heterogeneous, which did not permit the use of a single formal critical appraisal assessment tool. The inclusion of multiple study designs was necessary to address the study questions, and care was taken in the study selection, as observed by Peinemann et al., who stated that a critical assessment of study quality is all the more important to contextualize the results in a systematic review [37].

We critically appraised each of the included studies on ten domains. We reported our assessment of the risk of bias (low, high, or unclear risk) for each domain, together with a descriptive summary of the information that influenced our judgement. A study was assigned a low risk of bias if it scored low bias in at least seven of the domains; a high risk of bias if it scored high risk in at least four domains; and a moderate if six domains scored low risk and not more than three high risk scores. Two review authors (CM and LK) applied the criteria independently, and a third review author (LA) arbitrated any disagreements. Of the sixteen studies, seven were thus classified as having a low-risk of bias, six were of moderate risk of bias, and three were of high risk of bias. The following Table 4 is a summary of the domain analysis:

**3.5.1 Random sequence generation (selection bias).** Nine studies assigned participants into intervention and control groups quasi-randomly by date of birth, day of visit, record number, or alternate allocation; and non-randomly based on participants' or clinician's choice, test results, and availability [19–25, 27, 34]. As such, they were deemed high-risk for random sequence generation. The rest (seven) followed randomisation and therefore were rated as low risk.

**3.5.2 Allocation concealment (selection bias).** Most of the studies prevented participants from predicting which intervention and control groups they would be allocated to and therefore were rated low-risk. Three studies did not conceal allocation [26, 30, 34] and were rated high-risk. One study was not very clear on allocation concealment [32] and therefore was rated unclear.

**3.5.3 Blinding of outcome assessment (detection bias).** Our analysis showed that only four studies had their measurement likely influenced by no blinding or broken blinding [19, 23, 24, 30] and therefore were rated high risk. The rest of the studies (twelve) without such influence were thus rated low risk of detection bias.

**3.5.4 Blinding of participants and personnel (performance bias).** We rated four studies as high risk [23, 27, 28, 30]. This was because there was either no blinding, incomplete blinding, or broken blinding, which would more likely have influenced the outcomes. In the other twelve studies, some were blinded, and despite no blinding or incomplete blinding in others, our analysis showed that there were no indications that these influenced the outcomes. As such, these twelve studies were rated as having low-risk of performance bias.

**Table 4. Summary of risk of bias assessments.**

| | Random Sequence generation (Selection bias) | Allocation concealment (Selection bias) | Blinding of outcome assessment (Detection bias) | Blinding of participants and personnel (Performance bias) | Incomplete outcome data (Attrition Bias) | Selective reporting (Reporting bias) | Baseline characteristics | Contamination | Incorrect analysis | Other biases |
|---|---|---|---|---|---|---|---|---|---|---|
| Zhou G et al., 2022 | Low | Low | Low | Low | Low | Low | Low | Unclear | Low | High |
| Mangani C. et al., 2021 | High | Low | High | Low | Unclear | Low | Low | Unclear | Low | High |
| Jaleta KT et al., 2013 | High | Low | Low | Low | Unclear | Low | Low | Low | Low | Low |
| Okech BA et al., 2008 | High | Low | Low | Low | Unclear | Low | Low | High | Low | High |
| Elmardi KA et al., 2021 | High | Low | Low | Low | High | Low | Low | High | Low | High |
| Kibret S et al., 2014 | Low | Low | Low | Low | Unclear | Low | Low | Unclear | Low | Low |
| Yohannes M et al., 2005 | High | Low | High | Low | Unclear | Low | Low | Low | Low | Low |
| Fillinger U et al., 2009 | High | Low | Low | High | Unclear | Low | Low | Low | Unclear | High |
| Majembere S et al., 2010 | Low | Low | Low | High | Unclear | Low | Low | Low | Unclear | Low |
| Shiff CJ et al., 1973 | High | Low | High | High | Unclear | Low | Low | Low | Unclear | High |
| Tameim O et al., 1985 | Low | Unclear | Low | Low | Unclear | High | Low | Low | Low | High |
| Trienekens SCM et al., 2022 | Low | High | High | High | Low | Low | Low | Low | Low | Low |
| Fenwick A and Jorgensen T. A., 1972 | High | Low | Low | Low | High | Low | Low | Unclear | Low | High |
| Sokolow s. et al. 2015 | Low | High | Low | Low | Low | Low | Low | Low | Low | Low |
| Lelo A et al., 2014 | Low | Low | Low | Low | Low | High | Low | Low | Low | Low |
| Nikiema A S et al. | High | High | Low | Low | Low | Low | Low | Unclear | Low | High |
| **Key** | | | | | | | | | | |
| Unclear | Unclear | | | | | | | | | |
| High | High | | | | | | | | | |
| Low | Low | | | | | | | | | |

**3.5.5 Incomplete outcome data (attrition bias).** The team also assessed the risk of bias by considering if they accounted for data for all participants who were recruited in the study at the end of the study. We reviewed if there were reported cases of loss to follow-up and intention-to-treat analysis. Five studies were rated low-risk [26, 29, 30, 33, 34]. The rest were rated as high-risk because outcome data for all participants was not available for our review.

**3.5.6 Selective reporting (reporting bias).** Most studies (fourteen) were rated low risk of selective reporting because their protocols were available (and even those whose protocols

were not available) and showed that they correctly reported on pre-specified and expected outcomes of interest. Only two did not follow such guidelines and were rated as high-risk [23, 32].

**3.5.7 Baseline characteristics.** All studies had baseline characteristics properly recorded at the start and end of the study. They were all, therefore, rated low risk.

**3.5.8 Contamination.** We rated two studies high on the risk of contamination [21, 22]. We rated five as unclear [19, 25, 29, 31, 34] and the rest were rated as low-risk.

**3.5.9 Incorrect analysis.** We rated only three studies unclear on incorrect analysis because some information was not available for our judgement [23, 27, 28]. The rest were rated low, as we ascertained that they had followed all the required steps in the data analysis.

**3.5.10 Other biases.** We rated six studies low-risk because there was information that they adjusted for confounders [20, 24, 26, 28, 31, 33]. The rest did not state so, and were thus rated as high-risk.

## 4. Discussion

### 4.1 Summary of main results

This systematic review synthesized evidence on VBD control interventions in agricultural and irrigation areas in SSA. The study found various VBD control methods used, including a combination of some of the interventions. Amongst them are IVM, LSM, MDA, LLIN, ITN, IRS, mollusciciding, biological control, and social behaviour change, which resulted in a reduction in vector densities, VBD risk, and VBD prevalence. Overall findings show the possibility of controlling VBDs in areas where irrigation agriculture is practiced using one or a combination of some of these interventions. With VBDs accounting for 17% of the global infectious disease burden, such evidence is needed to contribute to countries' and WHO's global VBD control efforts [2, 11].

Our review found that IVM in irrigation areas reported remarkable success in VBD control and management. Elsewhere, some studies of IVM reported no improved outcomes. For example, hotspot targeting of interventions in the Western Kenyan highlands failed to reduce malaria transmission [38] and the addition of early-season larval control and IRS to high ITN coverage in Kenya indicated a limited added impact on reducing malaria transmission. the addition of early-season larval control and IRS to high ITN coverage in Kenya indicated a limited added impact on reducing malaria transmission [39]. Our study, however, proved the contrary. Despite some studies reporting interventions within an IVM being comparatively more effective than others [22, 27], the complementarity in IVM demonstrated overall improvement of the success rates in VBD reduction not only in malaria [21, 22, 27] but also in schistosomiasis [25, 32] control and management. Several reviews agree with the positive outcomes, crediting IVM for the remarkable VBD reduction across the globe through reducing the vector population, vector breeding sites, the entomology index, and consequently VBD prevalence [40, 41], and adding community participation and capacity building as elements of its sustainability. These results, however, show that not every IVM intervention is guaranteed success, and therefore care has to be taken on which interventions to combine in order to balance improved outcomes with cost-effectiveness. Nevertheless, it has been shown that IVM largely resulted in improved health outcomes, which impacted positively on school attendance among school-going children and improved agricultural productivity in the irrigation schemes.

In this review, we have seen that LSM reported mixed results on VBD reduction in irrigation areas. LSM has been more effective in reducing vector abundance by targeting their aquatic stage; however, this has not automatically translated into VBD reduction in all cases. LSM has been noted to work mainly on malaria control and management, although with a low

degree of success when employed singularly. While LSM has been successful in reducing the pupal (aquatic) stage of the vector population, the outcomes have not been successful for adult vectors [28]. Elsewhere, bio-larviciding with Bti has been noted to work only on specific vectors (e.g., Anopheles gambiae larvae but not Culex quinquefasciatus) and therefore has a very limited impact on overall VBD control and management [42]. Such outcomes exclude LSM from becoming a core intervention in VBD control but remain an important VBD control intervention in irrigation areas where vector breeding sites proliferate. These results agree with other reviews [43, 44] and the WHO recommendations that LSM is suitable in various eco-epidemiological settings, but should be a supplementary measure alongside the core interventions [45]. In addition, for better larvicidal initiatives and success rates, a similar study done in Burkina Faso recommended taking into consideration some common factors, such as strong local partnerships, meticulous planning, and reliable funding [46].

IRS and LLIN are some of the individual VBD control and management interventions in irrigation areas targeting malaria that registered promising results in the study. Despite IRS' effectiveness being compromised by all-year-round irrigation practices like sugarcane irrigation farming, it was still considerably highly effective [29]. This concurs with other studies attributing high IRS coverage to reduced malaria prevalence in Africa regardless of whether they used pyrethroids [47] or non-pyrethroid-like [48] insecticides. Similarly, high LLIN coverage has been notably effective in reducing malaria cases in irrigation schemes, with the effectiveness becoming higher in communities located further away from the irrigation schemes [19].

Despite MDA registering mixed results in this review, where it was more effective with onchocerciasis in Burkina Faso [34], but no significant change was registered in the schistosome population in Kenya [33], other studies elsewhere reported positive results. For example, a five-year annual MDA programme recorded a significant reduction of onchocerciasis microfilaremia (MF) prevalence in Sierra Leone [49] and when coverage was extended, an MDA intervention on schistosomiasis recorded effectiveness in communities and schools in Malawi [50]. This shows that with strategic population targeting, MDA could be effective in reducing VBD prevalence.

Mollusciciding, social behavioural change, and biological control have also recorded positive impacts on schistosomiasis reduction. The effectiveness of mollusciciding in reducing schistosomiasis prevalence by over 50% on irrigation farms in Zimbabwe resulted in labour efficiency, which ultimately improved agricultural productivity [23]. A systematic review of the effectiveness of mollusciciding in Africa echoes that it is effective in reducing schistosoma over time, leading to greater impact [51]. Despite a review of behavioural change interventions for control and elimination of schistosomiasis finding that their effectiveness is elusive due to varied indicators used to measure risk behaviours [52], our study found that employing behavioural change practices that reduced water contact behaviour amongst school-going children in Uganda registered schistosoma reduction, impacting positively on their school attendance [30]. Biological control of S. hematobium using native prawns was also deemed successful in reducing schistosomiasis prevalence [26]. This is in line with a review of several biological control measures for schistosomiasis transmission, which found that, unlike other interventions, biological controls are inexpensive, environmentally friendly, and non-toxic to many non-target organisms [53]. These results present several interventions applicable to schistosomiasis control and management in irrigation areas.

Among the notable behavioural changes reported in this review as a result of VBD control and management interventions are increased ITN usage [21, 27], improved health services-seeking behaviours such as the use of other combinations of malaria control interventions [27], and improved WASH practices [30, 33]. This was similar to other studies, which

registered increased demand for and uptake of health services by the communities in Malawi [54] and an improved trend in malaria testing and treatment-seeking behaviour by communities from 74% at the start to 85.5% in Southwestern Ethiopia [55]. However, IVM in some malaria studies registered negative behaviour with ITN usage declining in communities that felt no importance of ITN/LLIN use in houses that had undergone IRS [27, 56].

Our review showed that some VBD control and management interventions can be done cost-effectively. Amongst the VBD control interventions, available evidence identifies standard IRS as very costly considering its wide coverage, insecticides required, and implements used [56], but states that there is up to 55% cost savings if it is targeted IRS [57]. Our review found that, depending on the combination of interventions, IVM may be very cost-effective [32]. Another intervention found to be cost-effective is LSM, especially where it involves environmental modifications and targeted application [27], and if bio-larviciding uses alternative water dispensable formulations [57], or long-lasting formulations [58], or those like Bti, which may also be mixed with fertilizer and applied together as a routine husbandry practice without demanding additional labour [42].

The study revealed a wide range of intervention approaches that can be used in VBD control. However, most of them were not without challenges. Amongst the challenges noted in this review were: seasonal changes and climate variability, which influence vector breeding, posing a challenge to LSM; competing interests between humans and predators identified in biological control; and drug resistance in MDAs. Evidence beyond this review shows that chemical-based interventions are indeed challenged by insecticide resistance as a result of pesticide overuse, for example, in IRS and LLIN/ITN [39]. While the WHO has stepped in to regulate pesticide usage, the unavailability of alternative pesticides has posed yet another challenge to VBD control efforts [59]. A similar observation was made in this review, where it was noted that extending MDA to a larger population without consideration may lead to drug resistance due to selection pressure [33]. This calls for more studies on alternative pesticides and drugs to curb the growing threat of insecticide and drug resistance. Further, the perceived effectiveness of one intervention may lead to challenging behaviour when continuing with other interventions implemented within an IVM approach. A study outside this review done in Benin showed that while intermittent flooding as an LSM approach demonstrated great potential to reduce vector abundance, reduce malaria transmission, and increase rice yield, communities could not rapidly accept or easily switch to the new methods due to a lack of understanding and training in the method [60]. This places community behaviour as a challenge in VBD control interventions, but can be resolved through community education and mobilisation [55, 61, 62]. Notwithstanding the challenges, vector control remains the key and most effective tool in the fight against VBDs in SSA [63].

## 4.2 Potential biases in the review process

The validity of the results in this review may be limited by the conduct and reporting of the studies from which the data were extracted and pooled.

## 5. Authors' conclusions

### 5.1 Implications for practice

The review synthesised evidence on applicable VBD control interventions in agricultural and irrigation areas in sub-Saharan Africa. The study found various VBD control interventions across the region, including IRS, LLIN/ITN, LSM, MDA, IVM, and mollusciciding. IVM was the most commonly practiced, followed by LSM, MDA, mollusciciding, IRS, and ITN. IVM was the most successful due to the complementarity of the various interventions involved, which together

resulted in a reduction in vector density and a reduction in VBD prevalence. Though LSM initiatives were noted to be successful, they were largely recommended as supplementary to IRS and ITN, with strong local partnerships, meticulous planning, and reliable funding for the initiatives. MDA was very successful when the population was strategically targeted to avoid creating a drug resistance problem. IRS has been notably highly effective with both the use of pyrethroids and non-pyrethroid-like insecticides. The success of these various interventions has shown that it is practical to control and manage VBDs in areas where irrigation is in practice, leading to improved health among communities living in irrigation areas as well as increased agricultural productivity. However, the major challenges are seasonal changes and climate variability, insecticide and drug resistance, and farmers' attitudes toward accepting the interventions.

## 5.2 Implications for research

Most of the evidence on VBD control interventions in agricultural and irrigation areas in sub-Saharan Africa came from malaria studies, with a few from schistosomiasis and onchocerciasis and nothing from other VBDs. The deficiency in evidence of other VBDs therefore poses a challenge to rigorous assessment of their applicability in farming and irrigated areas and therefore calls for more studies on the same in sub-Saharan Africa.

## Supporting information

**S1 Appendix. Summary characteristics of excluded studies.**
(DOCX)

**S2 Appendix. Search strategies.**
(DOCX)

**S3 Appendix. PRISMA 2020 checklist.**
(DOCX)

## Author Contributions

**Conceptualization:** Rose Oronje.

**Data curation:** Levi Kalitsilo, Leila Abdullahi, Lily Mwandira, Collins Mitambo.

**Formal analysis:** Levi Kalitsilo, Leila Abdullahi, Hleziwe Hara, Collins Mitambo.

**Funding acquisition:** Rose Oronje.

**Investigation:** Levi Kalitsilo, Lily Mwandira.

**Methodology:** Levi Kalitsilo, Lily Mwandira, Rose Oronje.

**Project administration:** Rose Oronje.

**Resources:** Rose Oronje.

**Supervision:** Nyanyiwe Mbeye, Rose Oronje.

**Validation:** Levi Kalitsilo, Leila Abdullahi, Nyanyiwe Mbeye, Hleziwe Hara, Collins Mitambo.

**Visualization:** Collins Mitambo.

**Writing – original draft:** Levi Kalitsilo.

**Writing – review & editing:** Levi Kalitsilo, Leila Abdullahi, Nyanyiwe Mbeye, Hleziwe Hara, Collins Mitambo, Rose Oronje.

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
