## [Decision Letter · Decision Letter 0]

10 Mar 2024

PONE-D-24-02586Vector-Borne Disease Control Interventions in Agricultural and Irrigation Areas in Sub-Saharan Africa: A Systematic ReviewPLOS ONE

Dear Dr. Kalitsilo,

Thank you for submitting your manuscript to PLOS ONE. After careful consideration, we feel that it has merit but does not fully meet PLOS ONE’s publication criteria as it currently stands. Therefore, we invite you to submit a revised version of the manuscript that addresses the points raised during the review process.

We look forward to receiving your revised manuscript.

Kind regards,

Sammy O. Sam-Wobo

Academic Editor

PLOS ONE

“We acknowledge NIHR funding to the study through the Shire-Vec Project.”

“NIHR funded the study through the Shire-Vec Project.”

Please include your amended statements within your cover letter; we will change the online submission form on your behalf."

4. We notice that your supplementary tables are included in the manuscript file. Please remove them and upload them with the file type 'Supporting Information'. Please ensure that each Supporting Information file has a legend listed in the manuscript after the references list.

Additional Editor Comments:

Authors to go through and effect the corrections as stated

Reviewers' comments:

Reviewer's Responses to Questions

**Comments to the Author**

1. Is the manuscript technically sound, and do the data support the conclusions?

Reviewer #1: Yes

2. Has the statistical analysis been performed appropriately and rigorously? 

Reviewer #1: Yes

3. Have the authors made all data underlying the findings in their manuscript fully available?

Reviewer #1: Yes

4. Is the manuscript presented in an intelligible fashion and written in standard English?

Reviewer #1: Yes

5. Review Comments to the Author

Reviewer #1: the manuscripts reviewed various vector borne diseases control interventions in agricultural and irrigation areas in ssa using research outputs. the authors did an extensive search for relevant publications on the topic. the article was intelligently written with good data analyses

6. PLOS authors have the option to publish the peer review history of their article (what does this mean?). If published, this will include your full peer review and any attached files.

Reviewer #1: **Yes: **Innocent Chukwuemeka Omalu

---

## [Author Response · Author response to Decision Letter 0]

26 Mar 2024

Below is a point-by-point response to the academic editor's and reviewer's comments. The responses are in italics.

Reviewer’s comment number 1:

Please ensure that your manuscript meets PLOS ONE’s style requirements, including those for file naming. The PLOS ONE style templates can be found at

Response: 

The title page has been edited accordingly. 

a. Corresponding author indicated with an Asterisk (*), line number 3.

b. Author affiliations indicated with superscripts, lines 5 to 8.

Reviewer’s comment number 2:

We note that the grant information you provided in the ‘Funding Information’ and ‘Financial Disclosure’ sections do not match.

When you resubmit, please ensure that you provide the correct grant numbers for the awards you received for your study in the Funding Information section.

Response: Funding details updated: grant numbers verified.

Reviewer’s comment number 3:

Thank you for stating the following in the Acknowledgments Section of your manuscript:

“We acknowledge NIHR funding to the study through the Shire-Vec Project.”

“NIHR funded the study through the Shire-Vec Project.”

Response: As instructed, the acknowledgement of financing has been removed from the manuscript. However, the acknowledgment should say as follows where it will be made;

“NIHR funded the study under grant number NIHR133144.”

Reviewer’s comment number 4:

We notice that your supplementary tables are included in the manuscript file. Please remove them and upload them with the file type Supporting Information. Please ensure that each Supporting Information file has a legend listed in the manuscript after the references list.

Response: Supplementary tables: Appendices S1 and S2 have been uploaded as supplementary documentation after being taken out of the manuscript.

Reviewer’s comment number 5:

Response: Reference list verified; it is accurate.

Additional Editor Comments 6:

Authors to go through and effect the corrections as stated

Response: Reviewed the manuscript and made the necessary revisions.

Reviewers comment number 7 on Data Availability Statement:

We note that your Data Availability Statement is currently as follows: [All relevant data are within the manuscript and its Supporting Information files.]

Response: I confirm that all the raw data required to replicate the results of your study (Qualitative) is within the manuscript and the supporting documents provided.

---

## [Editor Report · Decision Letter 1]

1 Apr 2024

Vector-Borne Disease Control Interventions in Agricultural and Irrigation Areas in Sub-Saharan Africa: A Systematic Review

PONE-D-24-02586R1

Dear Dr. Kalitsilo,

We’re pleased to inform you that your manuscript has been judged scientifically suitable for publication and will be formally accepted for publication once it meets all outstanding technical requirements.

Kind regards,

Sammy O. Sam-Wobo

Academic Editor

PLOS ONE
---

## [Editor Report · Acceptance letter]

5 Apr 2024

PONE-D-24-02586R1 

PLOS ONE

Dear Dr. Kalitsilo, 

I'm pleased to inform you that your manuscript has been deemed suitable for publication in PLOS ONE. Congratulations! Your manuscript is now being handed over to our production team.

Kind regards, 

on behalf of

Dr. Sammy O. Sam-Wobo 

Academic Editor

PLOS ONE